# Micro Injection Molding of Drug-Loaded Round Window Niche Implants for an Animal Model Using 3D-Printed Molds

**DOI:** 10.3390/pharmaceutics15061584

**Published:** 2023-05-24

**Authors:** Robert Mau, Thomas Eickner, Gábor Jüttner, Ziwen Gao, Chunjiang Wei, Nicklas Fiedler, Volkmar Senz, Thomas Lenarz, Niels Grabow, Verena Scheper, Hermann Seitz

**Affiliations:** 1Microfluidics, Faculty of Mechanical Engineering and Marine Technology, University of Rostock, Justus-von-Liebig Weg 6, 18059 Rostock, Germany; robert.mau@uni-rostock.de; 2Institute for Biomedical Engineering, University Medical Center Rostock, Friedrich-Barnewitz-Straße 4, 18119 Rostock, Germany; thomas.eickner@uni-rostock.de (T.E.); nicklas.fiedler@uni-rostock.de (N.F.); volkmar.senz@uni-rostock.de (V.S.); niels.grabow@uni-rostock.de (N.G.); 3Kunststoff-Zentrum in Leipzig gGmbH (KUZ), Erich-Zeigner-Allee 44, 04229 Leipzig, Germany; juettner@kuz-leipzig.de; 4Lower Saxony Center for Biomedical Engineering, Implant Research and Development (NIFE), Department of Otorhinolaryngology, Head and Neck Surgery, Hannover Medical School, Stadtfelddamm 34, 30625 Hannover, Germany; gao.ziwen@mh-hannover.de (Z.G.); wei.chunjiang@mh-hannover.de (C.W.); lenarz.thomas@mh-hannover.de (T.L.); 5Cluster of Excellence “Hearing4all”, Department of Otorhinolaryngology, Head and Neck Surgery, Hannover Medical School, Carl-Neuberg-Straße 1, 30625 Hannover, Germany; 6Department Life, Light & Matter, Interdisciplinary Faculty, University of Rostock, Albert-Einstein-Str. 25, 18059 Rostock, Germany

**Keywords:** micro injection molding, 3D printing, rapid tooling, digital light processing, implant, drug delivery system, dexamethasone, anti-inflammatory, TNF-α, biocompatibility, inner ear therapy

## Abstract

A novel approach for the long-term medical treatment of the inner ear is the diffusion of drugs through the round window membrane from a patient-individualized, drug-eluting implant, which is inserted in the middle ear. In this study, drug-loaded (10 wt% Dexamethasone) guinea pig round window niche implants (GP-RNIs, ~1.30 mm × 0.95 mm × 0.60 mm) were manufactured with high precision via micro injection molding (µIM, T_mold_ = 160 °C, crosslinking time of 120 s). Each implant has a handle (~3.00 mm × 1.00 mm × 0.30 mm) that can be used to hold the implant. A medical-grade silicone elastomer was used as implant material. Molds for µIM were 3D printed from a commercially available resin (T_G_ = 84 °C) via a high-resolution DLP process (xy resolution of 32 µm, z resolution of 10 µm, 3D printing time of about 6 h). Drug release, biocompatibility, and bioefficacy of the GP-RNIs were investigated in vitro. GP-RNIs could be successfully produced. The wear of the molds due to thermal stress was observed. However, the molds are suitable for single use in the µIM process. About 10% of the drug load (8.2 ± 0.6 µg) was released after 6 weeks (medium: isotonic saline). The implants showed high biocompatibility over 28 days (lowest cell viability ~80%). Moreover, we found anti-inflammatory effects over 28 days in a TNF-α-reduction test. These results are promising for the development of long-term drug-releasing implants for human inner ear therapy.

## 1. Introduction

There is increasing interest in novel concepts of medical treatment of the inner ear in order to treat disorders such as Menière’s disease (MD) and idiopathic sudden sensorineural hearing loss (ISSHL). Pharmaceutical substances can pass from the middle to the inner ear via diffusion through the semipermeable round window membrane (RWM). The RWM is located deep in a recess, the round window niche (RWN), between the middle and inner ear (Figure 1). There are various pharmacological treatment methods and drug delivery strategies. For example, it is a common treatment to inject drugs or drug-laden gels directly into the middle ear cavity (intratympanically) by needle through the tympanic membrane [1]. However, such methods have a significant disadvantage. Large portions of the applied drug cannot diffuse through the RWM in the inner ear because it does not come into sufficient contact with it. Instead, much of the applied drug is absorbed by the mucosa of the middle ear or evacuated from the middle ear space by the Eustachian tube [2,3].

To overcome this drawback and to provide a more efficient and safe administration route for controlled drug release in the inner ear, we introduced a new concept of a patient-individualized, drug-loaded round window niche implant (RNI) [4,5]. Following our concept, an improved drug transport into the inner ear for several weeks might be obtained via drug diffusion from an RNI through the RWM. For that purpose, an RNI must meet the individual anatomical needs of a patient to fit precisely onto the RWM, and it should have dimensions of just a few millimeters in xyz directions. The RNI should be characterized by a soft and stretchable mechanical behavior, as we have already found in a prototype implantation study [5]. Moreover, inflammation-suppressing substances, such as glucocorticoids and especially dexamethasone (DEX), are promising for the drug load of an RNI. These substances have been proven to be promising in treating inner ear pathologies, including sudden sensorineural hearing loss (SSNHL) [6,7,8,9], Menière’s disease [10,11,12,13] and acute tinnitus [14]. Moreover, DEX positively affects the preservation of residual hearing and the reduction of fibrosis after cochlear implant (CI) surgery [15]. Recently, we found preferable concentration ranges for different DEX formulations to ensure biocompatibility and bioefficacy [16].

For highly individualized and complex products, such as the RNI described, 3D printing technology (also referred to as additive manufacturing) offers promising opportunities for both time- and cost-efficient production. In general, additive manufacturing methods that use photopolymerization enable the highest resolutions in the 3D printing sector [17]. There are various 3D printing processes that use photopolymerization, such as the vat photopolymerization methods (e.g., Digital Light Processing (DLP), stereolithography (SLA), two-photon polymerization (2PP)) or the material jetting methods (e.g., PolyJet, Multi-Jet Modeling (MJM)). The technology of DLP 3D printing is one of the most widely used processes because it offers high printing speeds and low running costs [18,19]. However, the availability and validity of ready-to-use, medical-grade materials that can be processed with photopolymerizing additive manufacturing methods are still limiting factors. On the one hand, there is a range of commercially available biocompatible photosensitive resins, but the majority of these are for dental applications (e.g., surgical guides, retainers, aligners, temporary dentures) or hearing aids and are suitable only for temporary skin contact (30+ days) or short-term mucosal contact (not more than 24 h) [20,21]. On the other hand, there is a lack of bioresins for applications such as tissue-engineered human constructs [21]. Very few resins certified as hemocompatible are commercially available. In a recent review article on biocompatible 3D printing resins, Guttridge et al. report that they found only one material (PrintoDent GR-20, pro3dure medical GmbH, Iserlohn, Germany) that was tested and certified as hemocompatible according to ISO 10993-4 [20]. When dealing with photopolymers, in general, there are risks such as cytotoxic, mutagenic and allergic reactions resulting from incomplete photopolymerization [18]. Information about the intended use, certification and postprocessing is highly variable for commercially available photopolymers [20]. It is often necessary to develop a specific postprocessing treatment to ensure sufficient biocompatibility for the desired use [20]. Another limitation is the lack of biodegradable resins. As described by Bao et al., there has been considerable progress in the development of biodegradable medical devices or implants using vat photopolymerization techniques such as DLP. However, further advancements in both novel materials and photopolymerization 3D printing techniques are needed for the challenging translation process toward clinical applications [22]. In contrast, conventional and well-established manufacturing via injection molding (IM) technologies is suitable for various materials, e.g., composite materials, foamed materials, thermoplastic and thermosetting plastics, rubber and even metals [23,24,25]. The use of medical-grade polymers [26,27] and the applicability of IM for the manufacturing of drug delivery systems have been investigated and established for many years [28]. Moreover, the micro injection molding method (µIM) enables the highest resolution and precision [29,30,31].

The processes of IM and µIM are based on the utilization of molds. Conventional manufacturing of metal molds via milling is relatively cost- and time-consuming, especially when there is a need for complex geometries. Therefore, IM and µIM are not usually used to manufacture a small series or even individualized single parts. To address this limitation, 3D printing technology enables rapid tooling as a cost- and time-saving method for mold manufacturing [32,33]. High-resolution photopolymerizing 3D printing methods, such as material jetting, SLA, or DLP, enable high process resolution for rapid tooling applications in micromanufacturing applications [34]. For instance, DLP potentially enables a cost reduction of 80% to 90% compared to conventional mold manufacturing [32].

In this study, we demonstrate a promising way of saving costs and time with a high-precision µIM of individualized DEX-loaded RNI using a medical-grade soft material. For that purpose, a mean guinea pig round window niche implant (GP-RNI) is manufactured via µIM. With a view to future investigation and translation, the GP-RNI features an exemplary implant geometry suitable for a favored animal model. We use high-resolution DLP 3D printing for rapid tooling of molds for the µIM process. After manufacturing the drug release, the biocompatibility and bioefficacy of the GP-RNI are investigated. Our manufacturing process aims to combine the best of both worlds of 3D printing and µIM: a cost- and time-saving, high-precision rapid tooling of molds via high-resolution 3D printing and the accessibility of a wide range of (polymeric) medical-grade materials for implant manufacturing, as such materials are well-established for µIM.

## 2. Materials and Methods

### 2.1. Three-Dimensional Models of Guinea Pig Round Window Niche and Mold

Figure 2A shows a photograph of the anatomical structure of an exemplary guinea pig round window niche and Figure 2B illustrates a graphic of the digital 3D model of a mean GP-RNI. The model has approximate nominal dimensions of a length of 1.30 mm, a width of 0.95 mm and a height of 0.60 mm. The digital 3D model was established by reconstructing 3D volumes from microCT images (XtremeCTII, ScancoMedical AG, Brüttisellen, Switzerland) via 3D Slicer^TM^ software version 4.11 (Surgical Planning Laboratory, Brigham and Women’s Hospital, Harvard Medical School, Boston, MA, USA) [35,36].

The 3D model of the mean GP-RNI was joined with a 3D model of a handle structure (Figure 3A (scheme), Figure 3B (final design)). The handle is an element to ensure good gripping and handling by forceps during the implantation process. This completed 3D model of the mean GP-RNI with a handle is designed to be manufactured via µIM. A sprue structure was added to the 3D model to obtain a flow path for the µIM material (Figure 3C). The sprue structure is removed from the µIM implant after the µIM process. Figure 3D shows the final 3D model of the mold (2 parts). It is based on the 3D model of the GP-RNI with a handle and the sprue. The 3D model of the mold will be used for rapid tooling of the mold via DLP 3D printing.

### 2.2. Rapid Tooling of Molds via Digital Light Processing

The 3D printing of the micro injection molds was performed with an Asiga Pro 4K45 (Asiga, Alexandria, Australia) using DLP technology. The xy resolution of the LED projector (UV light, λ = 385 nm, 4k-resolution-mode) was 32 µm and a build platform of 122 mm × 68 mm was installed. For the printing process, the photopolymeric resin Asiga PlasGRAY V2 (Asiga, Alexandria, Australia) [37] was used. The resin enabled a minimum layer thickness of 10 µm and enabled cured parts with a shore hardness of Shore D 82 and a glass transition temperature of T_G_ = 84 °C. It is designed for the manufacturing of highly detailed parts for dental, jewelry and design industries with high surface smoothness and quality. There were no reinforcing fillers, e.g., to enhance heat resistance for increased IM suitability, as the PlasGRAY V2 resin is a general purpose photopolymeric resin in DLP 3D printing.

Since a single mold is made of two parts (a lower half and an upper half), the parts of the same kind were 3D printed simultaneously. A total of 52 molds were manufactured. For that purpose, 52 lower halves of the molds (including supporting structure) were 3D printed simultaneously in 201 min in a first printing sequence. In addition, 52 upper halves of the molds (including the supporting structure) were 3D printed simultaneously in 166 min in a second printing sequence. All parts were built in a horizontal position (Figure 4). A base plate with a height of 0.4 mm was used as a supporting structure to ensure sufficient adhesion of the 3D-printed part with the built platform. The base plate was built using a layer resolution in z direction (layer height) of 100 µm per layer, exposed for 23.871 s per layer. Further layers of the mold parts were built using a layer height of 10 µm per layer, exposed for 0.498 s per layer. There was a material consumption of 0.417 mL per single mold.

Postprocessing steps were the washing of the printed parts in 98% isopropyl alcohol in an ultrasonic bath twice for 5 min, drying at room temperature for 30 min, and postcuring the mold halves for 2000 flashes on each side (total of 2 × 2000 flashes) in a UV curing unit Otoflash G171 (NK Optik GmbH, Baierbrunn, Germany).

### 2.3. Micro Injection Molding of Drug-Loaded Implants

For µIM investigations, a homogenous mixture of medical-grade silicone elastomer MED-4244 (NuSil Technology LLC, Radnor, PA, USA) containing 10 wt% DEX (powder, Sanofi SA, Paris, France) was prepared by manually stirring using a stainless steel laboratory scoop. MED-4244 is a two-part (10:1 *w*:*w*, part A: part B), pourable, translucent silicone elastomer and cures by heat via addition-cure chemistry. It can be used for implants that remain in the human body for a period of more than 29 days [38]. The stirring process was performed for several minutes until a uniform whitish coloration of the mixture appeared as an indicator of the homogenous distribution of the DEX powder in the liquid silicone. The homogeneity of the mixture and the absence of air bubbles in the mixture were evaluated by the naked eye. The preparation of the mixture was performed under an ambient atmosphere.

Micro injection molding (µIM) investigations were performed using a machine of type formicaPlast (Klöckner DESMA Elastomertechnik GmbH, Achim, Germany), modified as shown previously [29] and featuring an injection piston with a diameter of 3 mm. For µIM, a 3D-printed mold was inserted in the µIM machine using customized metallic housing as shown in Figure 5. Each mold was used only once for µIM. The µIM process was performed at a mold temperature of 160 °C and an injection flow rate of 4.2 mm^3^/s. The mold temperature of 160 °C was necessary for the heat-driven curing process of the used silicone elastomer. The shot weight (inclusive sprue) was 8.2 mm^3^. No extra holding pressure was applied. After injection of the prepared mixture of the silicone elastomer DEX mixture, there was a crosslinking time of 120 s.

### 2.4. Drug Release

Drug release was investigated via high-performance liquid chromatography (HPLC). For that purpose, the masses of the GP-RNI (n = 3) were determined on a Kern 770 microbalance (KERN & Sohn, Balingen, Germany), as shown in Table 1.

The GP-RNIs were placed in 4 mL glass vials and stored at 37 °C in 2 mL isotonic saline (B.Braun, Melsungen, Germany) on a lab shaker (Heidolph, Schwabach, Germany) at 100 rpm. For sampling, the medium was exchanged completely after defined time periods of 0.25; 0.75; 1.5; 3; 6; 13; 24; 29; 101; 197 and 317 h and then every 7 days for an additional 6 weeks. The medium was subsequently mixed 1:1 (*v*:*v*) with methanol (Carl Roth, Karlsruhe, Germany) and distilled water (Ultrapure water system (Sartorius, Göttingen, Germany)) prior to the HPLC measurements. Quantification of DEX was performed on a HPLC system (Knauer Wissenschaftlicher Gerätebau Dr. Ing. Herbert Knauer GmbH, Berlin, Germany) equipped with a Chromolith FastGrad RP-18e 50-2 column (Merck KGaA, Darmstadt, Germany). Methanol/Water 1:1 was used as the mobile phase in an isocratic chromatographic method at a flow rate of 0.8 mL/min. Detection occurred with a UV-Detector at the wavelength λ = 254 nm [39]. For calibration, DEX standards with concentrations of 0.1, 0.5, 1.0, 2.0, 5.0, 10 and 50 µg/mL were used.

### 2.5. Biocompatibility

For in vitro biocompatibility and bioefficacy (see Section 2.6.) studies, eluates were generated by incubating RNIs in 24-well plates (Nunc, Thermo Fisher Scientific, Waltham, MA, USA) with 500 µL saline (600 µL NaCl 0.9%, B. Braun, Melsungen, Germany) per well in an incubator (CB150; Binder, Tübingen, Germany; 37 °C, 5% CO_2_, 95% humidity) for 1, 3, 7, 10, 14, 21 and 28 days. On the day of collecting the supernatant, all of the supernatants in each well was taken out and separated into two Eppendorf tubes, one for a biocompatibility test and the second for a bioefficacy test. The wells were refilled with fresh 500 µL saline and the sample was further incubated until the next sampling time point was reached. The supernatants were stored at −20 °C before processing. All experiments were performed in triplicate and repeated three times.

A 3-(4,5-dimethylthiazol-2-yl)-2,5-diphenyltetrazolium bromide (MTT) assay (PanReac AppliChem, Darmstadt, Germany) was performed to investigate the biocompatibility of the supernatant as previously described [16]. NIH/3T3 fibroblasts (mouse, ATCC-Number: CRL-1658, German Collection of Microorganisms and Cell Cultures GmbH, Braunschweig, Germany; passage 3 to 10) were seeded in Dulbecco’s modified Eagle’s medium (DMEM, Bio and Sell GmbH, Feucht, Germany) with 10% fetal calf serum (FCS, Bio and Sell GmbH, Feucht, Germany), penicillin and streptomycin (100 units/mL each) in a humidified atmosphere (5% CO_2_/95% air, 37 °C) as shown before [16]. To perform the MTT assay, the fibroblasts were seeded in 96-well plates at a concentration of 1.5 × 10^4^ cells/mL with 100 µL fresh culture medium. After an incubation time of 24 h, the culture medium was replaced by a fresh culture medium and culture supernatant of GP-RNI samples at a 1:1 ratio. Cells treated with 0.1% DMSO were used as a positive control (PC) for a toxic effect on the cells. The negative control cells (blank) were cultured in a pure complemented medium for regular cell proliferation (50 μL 0.9% NaCl + 50 μL cell culture medium). For validation that the experiments were performed successfully, the PC and blank conditions ran in parallel with every single experiment. All experiments were performed in duplicate and repeated three times.

After 24 h, the medium was removed, replaced by 50 µL 0.5 mg/mL MTT reagent, and incubated for two hours in a humidified atmosphere (5% CO_2_/95% air, 37 °C). Subsequently, the MTT reagent medium was removed and replaced by 100 µL MTT solution (isopropanol) per well. To dissolve the formazan produced by MTT reduction, the MTT reagent medium was incubated for five minutes on a rotary shaker at room temperature of 21 °C. The optical density (OD) was determined at a wavelength of 570 nm utilizing a microplate reader (Gen5 2.06.Ink, BioTek Synergy™ H1HyBrid Reader, Santa Clara, CA, USA). The measurement of empty wells without cells was performed for a correction of the OD. The relative cell viability was calculated in percentage terms by dividing the empty-subtracted OD of the test groups by the empty-subtracted OD of the blank and multiplying the result by 100. Cell viability below 70% was judged as being cytotoxic, which is in accordance with ISO guideline 10993-5:2009 for the biological evaluation of medical devices. The normal distribution of data was checked (Kolmogorov–Smirnov Test) and an analysis of variance (ANOVA) and a Dunnett’s Multiple Comparison Test was conducted using GraphPad Prism^®^ version 8.4.3 (GraphPad Prism Software Inc., La Jolla, CA, USA). Means ± standard deviations of the data were reported and the statistical significance was considered at *p*-values less than 0.05.

### 2.6. Bioefficacy

A TNF-α-reduction test was performed for the investigation of the anti-inflammatory effect of the µIM-manufactured GP-RNI samples containing 10 wt% DEX as shown before [40]. It is assumed that the GP-RNI should release DEX into the supernatant (see Section 2.5. for supernatant sampling), and to be bioeffective, the released DEX should reduce the TNF-α-production of cells being stressed with lipopolysaccharide (LPS, Sigma-Aldrich, St. Louis, MO, USA). Experiments were performed in triplicate (n = 3) per plate (N = 3) for every condition: DC2.4 mouse cells (DCs) (Sigma-Aldrich, St. Louis, MO, USA, LOT:3093896) were cultured in 48-well plates in RPMI 1640 medium (Sigma-Aldrich, St. Louis, MO, USA), which was supplemented with non-essential amino acids (1 mmol/L, Sigma-Aldrich, St Louis, MO, USA) and 10% FCS (Bio & Sell GmbH, Feucht, Germany). The cells were cultivated for 24 h in an incubator and subsequently divided into negative control (NC), positive control (PC) and supernatant groups. All cells, except the NC, were stressed by adding 100 µL LPS to the medium (0.5 µg/mL). The PC and NC conditions were conducted in parallel with each single experiment for the validation that each experiment was performed successfully. The supernatants of the cultured GP-RNIs were added to the wells and the cells were incubated for an additional 24 h. After 24 h, the supernatant was collected and ELISA analysis was performed. ELISA kits (Boster Biological Technology, Pleasanton, CA, USA) were used in accordance with the manufacturer’s instructions. Each supernatant was applied in dilution and as a replicate to the ELISA plate. The absorbance of OD was recorded at a wavelength of 450 nm utilizing a MicroPlate Reader (Gen5 2.06.Ink, BioTekSynergy™ H1HyBrid Reader, Santa Clara, CA, USA).

Because of the non-normal distribution of data (Kolmogorov–Smirnov Test), subsequently, each sampling time point was separately tested for relevant differences in TNF reduction compared to PC using the Mann–Whitney U Test. The Friedman test (*p* < 0.0001), followed by Dunn’s Multiple Comparison Test, was run to detect differences between the dependent variable of the different time points. The data are reported as mean ± standard deviation. Statistical significance was considered at *p*-values less than 0.05.

## 3. Results and Discussion

### 3.1. Rapid Tooling of Molds via Digital Light Processing

Figure 6 shows both parts of a DLP 3D-printed mold after 3D printing but before finishing the parts by milling off the remnants of the supporting structure (Figure 6A,B, marked via white arrows). The upper half of the mold (Figure 6A) features the geometry of the upper half of the GP-RNI and the sprue. The sprue is a relatively large channel through which the liquid polymer material enters the mold. Furthermore, the lower half of the mold (Figure 6B) features the geometry of the GP-RNI and the form of the handle structure of the GP-RNI. The handle structure is used as a runner structure for the transport of liquid polymer inside the mold. Moreover, the lower half of the mold features a “cold slug” structure for the µIM process. The forms of the halves of the GP-RNI are of high precision because of the high resolution of the utilized DLP 3D printing process featuring a relatively low z-layer height of 10 µm per layer (Figure 6C,D).

### 3.2. Micro Injection Molding of Implant Prototypes

Figure 7 shows the finished and assembled 3D-printed halves of the mold. The parts fit closely. There is minimal clearance between the upper and lower halves before they are used for the µIM injection molding process (see Figure 7A). The edges of the parts are flush with each other. After the µIM process (single use, Figure 7B), the clearance between the upper and the lower halves is decreased. The upper half of the mold shows a crack as a sign of wear.

Figure 8 shows a separated mold after a finished µIM process. The mixture of the silicone elastomer (MED-4244) and the glucocorticoid DEX (10 wt%) was successfully injected and cured. Despite the decreased clearance between the parts (compare Figure 7A,B), the two halves were not significantly merged and the separation of the halves was easy. However, there are significant signs of wear all over the surface of both parts of the mold. Nevertheless, the separation of the GP-RNI from the mold worked well and did not cause any damage to the µIM implant.

Low thermal conductivity, low heat resistance and, consequently, low durability and high cycle times are known limitations of molds manufactured from available photopolymeric resins [32,34]. In this study, the combination of thermal and mechanical stress during the µIM process is most likely the reason for the wear of the DLP 3D-printed molds, as these are significant factors [32]. The glass transition temperature (T_G_) of the photopolymer PlasGRAY V2, which was used for DLP 3D printing of the molds, is T_G_ = 84 °C [37]. The mold temperature during the performed µIM process was T_mold_ = 160 °C. When the mold temperature is similar to T_G_, the polymer material may become brittle and prone to failure [32]. Zink et al. recommend keeping the mold temperature below the glass transition temperature of the mold material [41]. Following Zink et al., the parameter of the mold temperature significantly affects the applicability of 3D-printed polymeric molds. The mold material’s mechanical properties are impaired as a function of temperature, especially when using temperatures above T_G_. In another study by Martinho et al., the authors recommend keeping the mold temperature below 15 °C above T_G_ of the 3D-printed photopolymer resin [42]. In our study, we used a relatively high T_mold_ = 160 °C for a mold material with a T_G_ = 84 °C. The manufactured GP-RNIs were of proper quality. However, the molds showed significant wear because of thermal stress. Because of that, we used a single mold only once for our µIM process.

Due to economical manufacturing criteria, 3D-printed polymer molds can be well suited for low- and medium-volume injection molding production [32]. Commonly, for a medical implant, it is desirable to meet a patient’s individual needs. Implants such as the described RNI benefit from high customization. Low durability of the molds and relatively high cycle times of an injection molding process might be acceptable. As a result, the described rapid tooling process chain based on 3D-printed molds and µIM has proven to be a promising manufacturing technique for the production of highly individualized single parts. Furthermore, there are promising strategies to overcome the limitations of heat transport and heat resistance, such as composite materials [32] and innovative cooling channels in the 3D-printed molds [41,43]. Even relatively simple methods such as tight-fitting metal mold holders can help to deal with low heat resistance of 3D-printed molds, as we showed in [44].

Figure 9A shows the top view and Figure 9B shows the bottom view of two GP-RNI. The implants are homogeneously colored and show no failures such as burns, black spots, short shots or deformations. There are a few flash failures around the contour of the implant, where the mold halves met (Figure 9C). Moreover, the implant’s surface shows stair casing (Figure 9C) because the built resolution of the implant is limited to the DLP 3D-printing resolution of z = 10 µm per layer, which was used for mold manufacturing. Nevertheless, a layer height of z = 10 µm is a relatively low value compared to other methods, especially non-photopolymerzing 3D-printing methods [34]. Consequently, the utilized DLP technique allows µIM of implants with a relatively high resolution. An alternative photopolymerizing 3D-printing technique that enables a higher resolution and a lower staircase effect could be two-photon polymerization (2PP) [21]. However, in comparison to DLP, the use of 2PP would most likely lead to significantly higher 3D printing times. With our current state of knowledge, it is not clear which resolution is needed for therapeutically effective RNIs. In [45], we reported good fitting accuracy of prototypes of human RNIs with a z = 100 µm per layer, which were implanted in human cadaver RWN. We further found that a higher resolution, respectively a lower z-value per layer, leads to increased contour accuracy. The highest possible contour accuracy might be desirable. Increased contour accuracy most likely may lead to a better interface between the implant and the RWM. Consequently, a more effective drug transport into the inner ear by drug diffusion through the RWN might be achieved. Further investigations are needed at this point. At this point, the xy resolution of 32 µm and the z resolution of z = 10 µm (layer height) used here enable 3D printing of a relatively high resolution and a high contour accuracy.

We used the silicone elastomer MED-4244 since it is a soft and stretchable material [38]. Materials with such mechanical behavior are favorable for RNIs because of the beneficial tactile feedback and handling during implantation while also minimizing the likelihood of traumatizing sensitive structures such as the RWM during insertion [45]. Moreover, there is high potential to adapt our rapid tooling-based µIM manufacturing process to other medical-grade materials and biodegradable materials, as such materials are established for IM processes [26,27].

### 3.3. Drug Release

The release of DEX from the GP-RNI shows a two-phase progression with a burst release at the beginning, followed by a slower release (Figure 10A,B). The burst release occurred within the first 13 h (Figure 10C,D), during which a relatively large amount of DEX (about 0.7 µg in total) is released. The release of DEX from the GP-RNIs showed a diffusion-controlled mechanism, behaving like a matrix system [46,47]. As diffusion is dependent on the concentration gradient between the drug-releasing implant and the release medium, a faster diffusion results with a higher concentration gradient. The gradient remains maximal at the beginning of the release. A slower release phase follows with a linear slope. A release rate of about 1 µg/week is observed from the second week onwards. At the time of evaluation, this phase had not yet ended. After 6 weeks, about 10% of the DEX (8.2 ± 0.6 µg) had been released.

The drug release behavior found is promising, as the GP-RNI allows prolonged drug release over several weeks to months. Such long-release behavior can be beneficial compared to inner ear therapy methods [4]. Nevertheless, the drug release from GP-RNIs has to be tested in a more realistic scenario in vivo. By intratympanic application, the drug has to pass the RWM. This affects the drug concentration reached in the inner ear [48]. Further research on diffusion-based drug transport from the implant through the RWM is needed. Moreover, further research is needed to investigate whether the amount of DEX released is sufficient to achieve a positive therapeutic effect. It has been found in the literature that even relatively low DEX concentrations of 0.00118 mg/mL are effective [49]. However, the findings in the literature on therapeutic effective drug concentrations in cochlear pharmacotherapy are not consistent, and individual experimental parameters make it difficult to compare results [16]. In addition, the DEX concentrations must be chosen depending on the DEX formulation, as there may be significant differences in cytotoxicity [16].

The thermal stress from our µIM process might increase the risk of the degradation of the drug load in the processed material [48,50]. In our process, we used a mold temperature of T_mold_ = 160 °C and a crosslinking time of t = 2 min. These parameters should not lead to significant degradation of the drug load in the processed µIM material as DEX has a melting temperature of T = 262.4 °C. There is a rapid decomposition at higher temperatures [51], but process temperatures below are considered to be suitable. Farto-Vaamonde et al. successfully processed a DEX-loaded filament via extrusion-based 3D printing at extrusion temperatures of T = 220 °C [51]. In the work of Li et al., DEX was exposed to a temperature of T = 185 °C for a period of 5 min during hot melt extrusion, but the authors reported no significant drug degradation [52]. However, as Farto et al. recommend in [51], unnecessarily long heating periods at relatively high temperatures should be avoided as much as possible to prevent stability problems with the drug.

Further glucocorticoids, such as prednisone or hydrocortisone (cortisol), offer therapeutic potential for inner ear therapy [53] and are promising for inclusion in further investigations. Prednisone melts and degrades at temperatures of 230–235 °C [54]. For hydrocortisone, the start of thermal degradation was found at a temperature of 225 °C [55]. Therefore, both of these drugs could be suitable for our process. The risk of drug degradation due to thermal stress should be taken into account, especially when using further, thermal sensitive components. For example, proteins such as the growth factors brain-derived neurotrophic factor (BDNF) and insulinlike growth factor 1 (IGF1) have been identified as potentially protective for hearing [56]. They might be promising for long-term inner ear therapy. However, our process is not suitable for the processing of proteins, as they can denature at temperatures far below T = 160 °C.

### 3.4. Biocompatibility and Bioefficacy

The results of in vitro biocompatibility (cell viability) and bioefficacy (TNF-α-reduction test) investigations are shown in Figure 11. Compared to the blank (100%), the cell viability of the PC, including the cytotoxic agent DMSO, was significantly reduced (12 ± 9%; *p* < 0.001), proving the successful experimental setup. The cell viability of all GP-RNI-supernatant samples (mean ± SD; day 1: 114 ± 19%; day 3: 80 ± 13%; day 7: 94 ± 16%; day 10: 85 ± 22%; day 14: 99 ± 9%; day 21: 102 ± 9%; day 28: 104 ± 13%) did not differ significantly from that of the blank (Figure 11A). The lowest cell viability was 80.44 ± 13.30%. It was found when the supernatant of sampling day 3 was applied. It is still clearly above the 70% of the blank, which is the mark for indicating cytotoxic potential.

Cells without stress (NC) showed a very low basic level of TNF-α-production (11.58 ± 10.44 pg/mL), while this level significantly increased when LPS was added (PC, 776.2 ± 106 pg/mL). Compared to the PC, all tested supernatant reduced the TNF-α amount in the DC-cell-LPS-stress test significantly (Figure 11B). This anti-inflammatory effect was highest on the 10th day (188.2 ± 136.5 pg/mL) and lowest on the third day (318.9 ± 186.3 pg/mL). During the sampling period, the anti-inflammatory effect of the eluate varied. Data and results of statistical analyses are shown in the Appendix A (Table A1).

Our results show neither the usage of photopolymeric molds for µIM nor the drug load of DEX affect the biocompatibility of the used RNI material silicone elastomer MED-4244 critically. The exact amount of DEX released in the supernatant, which was used for biocompatibility investigations, is unknown at this point. Toxic effects of DEX are reported in the literature even for relatively low concentrations of 3 µM (0.00118 mg/mL, DEX: 392.46 g/mol) [49]. The authors report the start of toxic effects on outer hair cells by that drug concentration in vitro. However, as we reported previously [16], the findings in the literature concerning critical drug concentrations in cochlear pharmacotherapy are not consistent, or widespread, and are hard to compare as different individual experimental parameters must be considered. With an MTT assay, as we used here in this work, we found no significant toxic effects for DEX concentrations up to 2000 µM (0.784 mg/mL) [16]. With regard to the slow drug release behavior of the tested GP-RNI samples in isotonic saline, the DEX concentration in the supernatant should be far below 2000 µM. This supports our findings that the GP-RNIs containing DEX and that are made from 3D-printed molds are biocompatible.

We found significant anti-inflammatory effects during the whole 28-day course of the investigation. The anti-inflammatory effects of DEX are well known, and there are further potentials in terms of the protection of CI patients from hearing loss, fibrotic CI encapsulation and spiral ganglion degeneration [16]. In the literature, effective concentrations for DEX are found from 0.00118 mg/mL [49] to 24 mg/mL [57]. However, because of a wide variety of experimental parameters and treatment protocols, findings from the literature are hard to compare and there is a large variability between concentrations being toxic in vivo and those having a beneficial effect [16]. Further in vivo investigations need to show what DEX concentrations are needed in RNIs to receive specific therapeutic effects for inner ear therapy. Many studies highlight the therapeutic potentials of DEX for inner ear diseases, such as [15,58,59]. In addition to pure DEX, as we used here, there are other drug formulations, such as dexamethasone dihydrogen phosphate disodium (DPS). In a previous study, we found a slight tendency for DPS to be more effective in reducing TNF-α-production than other DEX formulations [16]. Moreover, other glucocorticoids, such as prednisone and hydrocortisone (cortisol), are promising for further investigations because they hold therapeutic potential for inner ear therapy [53].

## 4. Conclusions

We presented the high-precision manufacturing and analysis of drug-loaded implants for controlled drug delivery in the inner ear. In this study, mean guinea pig round window niche implants (GP-RNIs) were manufactured via micro injection molding (µIM) using molds manufactured via rapid tooling using a DLP 3D-printing process. A commercially available photopolymer resin was successfully used as the mold material. This photopolymer resin was primarily designed for highly detailed parts for dental, jewelry, and design. There was no need for reinforcing filler materials to improve material properties such as heat resistance. The 3D-printed molds were suitable for single use in our µIM process, which enables the individual manufacturing of highly patient-personalized implants.

A medical-grade silicone elastomer was drug-loaded with the glucocorticoid DEX (10 wt%) and successfully used for µIM of the implants. The µIM-manufactured implants showed high biocompatibility over a 28-day period. Moreover, we found anti-inflammatory effects over a 28-day period in a TNF-α-reduction test, which indicates high bioefficacy of the drug load. In vitro drug release investigations showed a burst release of about 0.7 µg DEX within the first 13 h in isotonic saline. A slower drug release phase follows with a linear slope. After 6 weeks, about 10% of the drug load of DEX (8.2 ± 0.6 µg) had been released. These results are promising for prolonged drug delivery of an RNI for inner ear therapy.

Further investigations will focus on in vivo testing of GP-RNIs. Drug transport through RWM in the inner ear and the therapeutic effect of GP-RNIs especially need to be examined. Moreover, further research is needed to adapt the presented rapid tooling-based µIM manufacturing process to other medical-grade materials, particularly biodegradable materials.

## Figures and Tables

**Figure 1 pharmaceutics-15-01584-f001:**
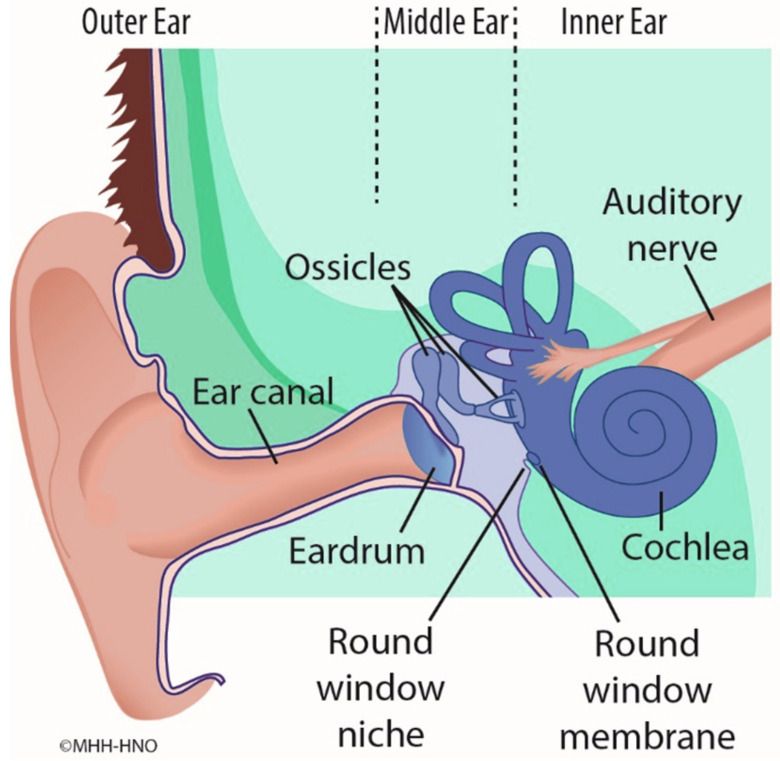
Scheme of the anatomic structures of outer ear, middle ear and inner ear. The round window membrane is located between the middle ear and the inner ear.

**Figure 2 pharmaceutics-15-01584-f002:**
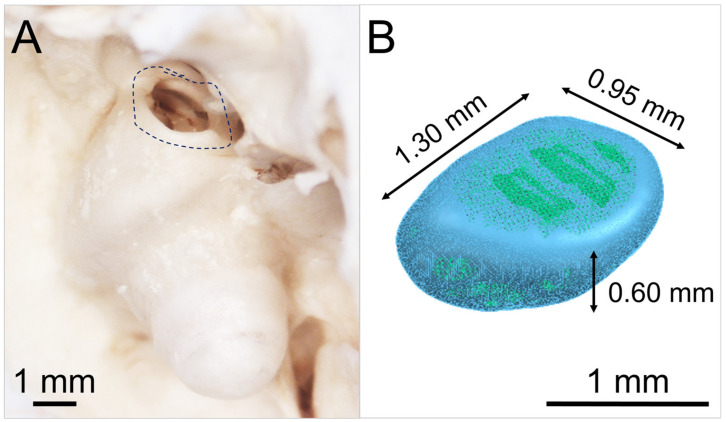
(**A**) Photograph of the anatomical structure of exemplary guinea pig round window niche (GP RWN, marked); (**B**) 3D model of mean guinea pig round window niche implant (GP-RNI). The approximate dimensions of the model are marked in the figure.

**Figure 3 pharmaceutics-15-01584-f003:**
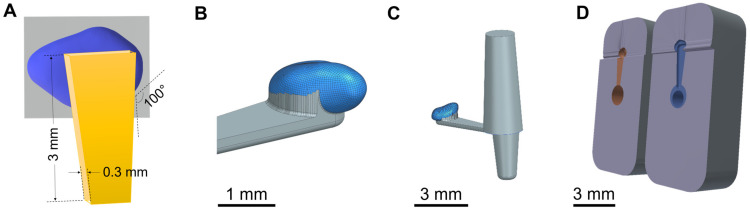
(**A**) Scheme of mean guinea pig round window niche implant (GP-RNI) (blue) with a handle (yellow, for a good gripping by forceps during the implantation process); (**B**) 3D model of mean GP-RNI (blue) with a handle (grey); (**C**) 3D model of GP-RNI (blue) with a handle for implantation (grey) and sprue for µIM (grey, conic); (**D**) 3D model of mold (2 parts, left: upper half, right: lower half) for µIM of GP-RNI with a handle (the negative halves of the implant with a handle and the sprue are colored orange and blue).

**Figure 4 pharmaceutics-15-01584-f004:**
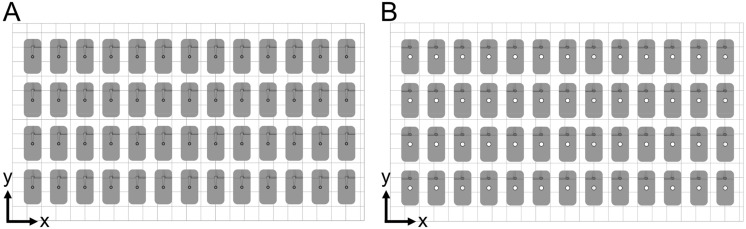
Placement of molds on build platform. Fifty-two of each lower (**A**) and upper (**B**) halves of the final molds were printed in the same orientation to eliminate possible differences in xy resolution.

**Figure 5 pharmaceutics-15-01584-f005:**
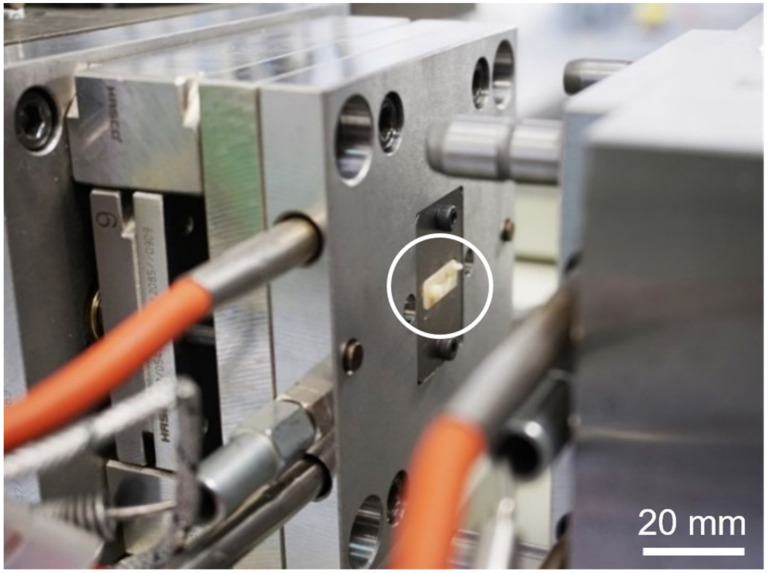
Micro injection molding (µIM) machine equipped with an exemplary 3D-printed mold (see white circle). A metallic holder keeps the mold in position.

**Figure 6 pharmaceutics-15-01584-f006:**
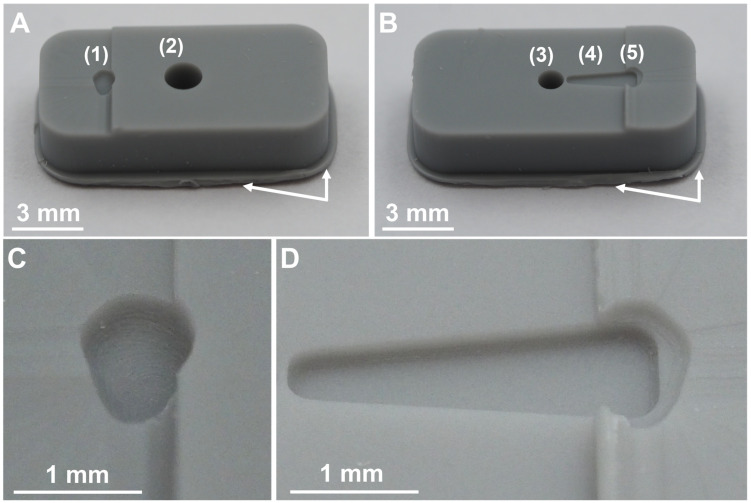
Photographs of DLP 3D-printed mold after 3D printing (but before finishing). The mold consists of two parts. (**A**) shows the upper half of mold featuring the form of the upper half of the GP-RNI (1) and the sprue (2). (**B**) shows the lower half of the mold featuring the form of the lower half of the GP-RNI (3) and the form of the handle structure of the GP-RNI (4). The mold is designed to use the handle structure as a runner structure for liquid µIM material during µIM process. Moreover, there is the “cold slug” structure (5) for µIM process. Remnants of the supporting structure are marked via white arrows. (**C**,**D**) show the forms of the GP-RNI in more detail. The small structures are 3D printed with a high grade of precision because a high-resolution DLP 3D printing process featuring a relatively low z-layer height of 10 µm per layer was used.

**Figure 7 pharmaceutics-15-01584-f007:**
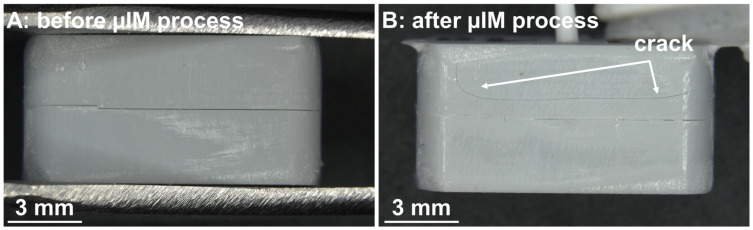
(**A**) Photograph of an assembled mold (made of upper and lower halves) before µIM process. Both parts of the mold fit well together. The edges of both halves are flush with each other. (**B**) A mold after µIM process. The upper half is cracked.

**Figure 8 pharmaceutics-15-01584-f008:**
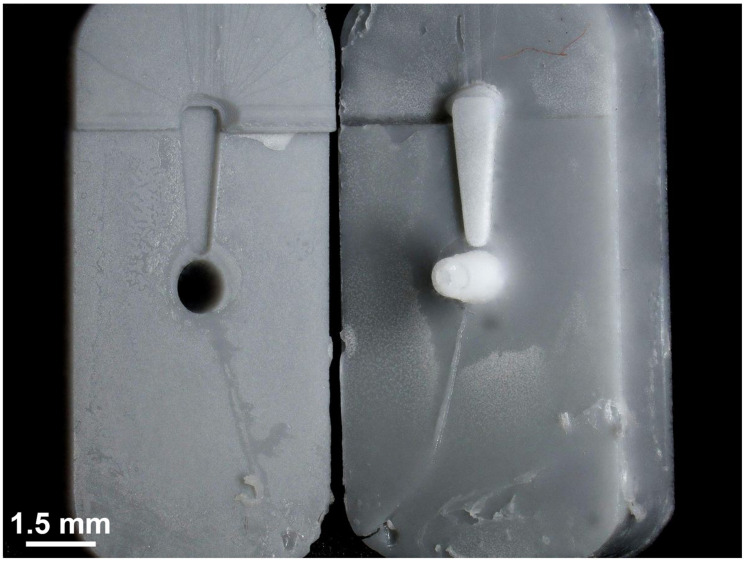
Photograph of a separated mold after the finished µIM process. The µIM material, a mixture of a medical-grade silicone elastomer (MED-4244, NuSil Technology) and DEX (10 wt%), was successfully injected and cured. It was separated from one part of the mold without any damage to the GP-RNI. There are significant signs of wear all over the surface of both parts of the mold.

**Figure 9 pharmaceutics-15-01584-f009:**
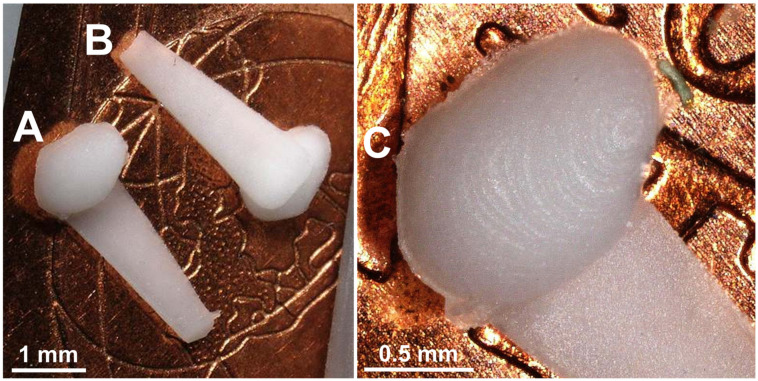
Photographs of two GP-RNI implants with handle structure manufactured via µIM under the usage of rapid tooling with high-resolution DLP 3D printing. (**A**) shows the top view and (**B**) shows the bottom view of the implant. The implants are manufactured from a medical-grade silicone elastomer (MED-4244) and are drug-loaded with the glucocorticoid DEX (10 wt%). The implants show no signs of burns, black spots, short shots or deformations. (**C**) marks a few flash failures around the implant contour where the mold halves met. Moreover, the implants show stair casing, as its resolution is limited to the DLP 3D printing z resolution of 10 µm/layer, which was used for mold manufacturing.

**Figure 10 pharmaceutics-15-01584-f010:**
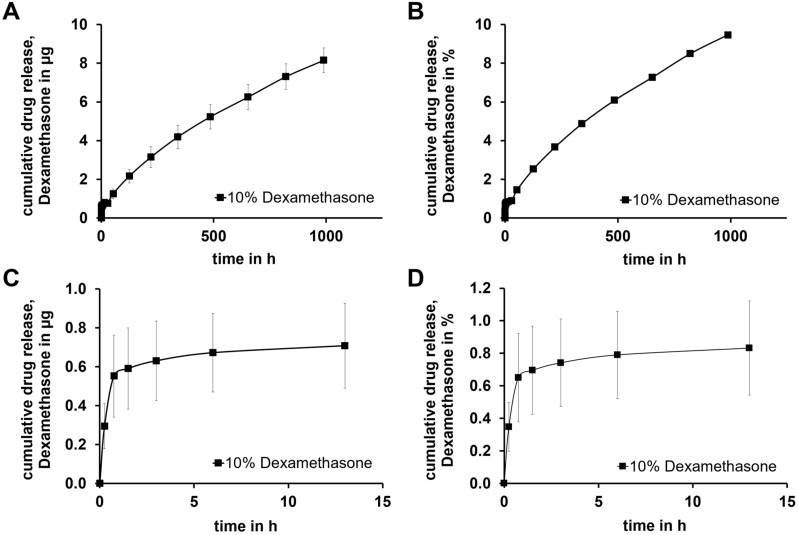
Diagrams show (**A**) cumulative drug release of DEX absolute amount of DEX for about 6 weeks, (**B**) relative DEX release for about 6 weeks, (**C**) enlarged representation of the complete release over the first 15 h, (**D**) enlarged representation of the relative release over the first 15 h; Diagrams (**B**,**D**) were normalized according to the calculated amount of DEX 10 % of the RNI-mass. Drug release testing continues.

**Figure 11 pharmaceutics-15-01584-f011:**
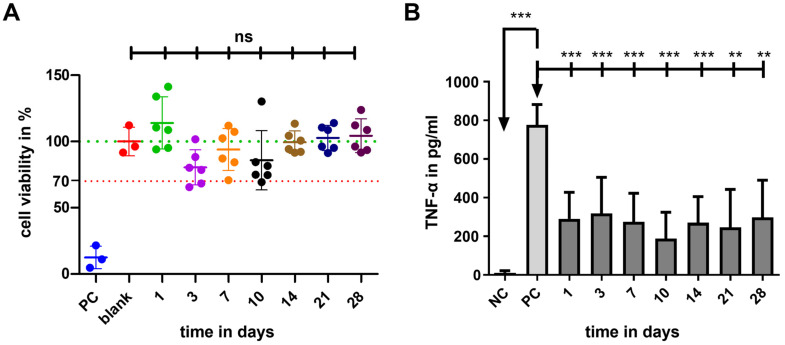
(**A**) Comparison of cell viability (CV in %) of fibroblasts treated with the supernatant of cultured GP-RNI sampled after various incubation times. The mean survival in blank is set as 100% (green line). PC illustrates the cytotoxicity of DMSO and the successful performance of the assay. The dotted line at 70% CV marks the toxicity level, based on the ISO guideline for biocompatibility testing of medical devices (ISO 10993-5:2009—8.5.1). The means of all data per time point were all in the safe range above 70% and statistical analysis did not report significant differences compared to the blank. Data are given as mean ± standard deviation with single experimental results included as dots (N = 3; n = 3); ns = not significant. (**B**) TNF-α amounts measured by ELISA in the supernatants of dendritic cells (DCs). TNF-α production is induced by the addition of 0.5 µg/mL LPS to the culture medium. This results in a high release of TNF-α in the PC when compared with the basic TNF-α level of unstressed cells in the NC. All tested RNI-eluates reduced the TNF-α amount in culture. Data are given as mean ± standard deviation and detected significances are marked with ** (*p* < 0.01) and *** (*p* < 0.001).

**Table 1 pharmaceutics-15-01584-t001:** Masses of the GP-RNIs prior to the drug release testing, resulting in a mean ± standard deviation of 0.86 ± 0.04 mg.

Sample of GP-RNIs	Mass in Mg
1	0.92
2	0.83
3	0.83

## Data Availability

The raw data required to reproduce the findings of this article are available from the corresponding authors upon reasonable request.

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
