# Peer review of "Micro Injection Molding of Drug-Loaded Round Window Niche Implants for an Animal Model Using 3D-Printed Molds"

_pharmaceutics, 2023, doi:10.3390/pharmaceutics15061584_

Round 1

Reviewer 1 Report

The manuscript entitled “Micro Injection Molding of Drug-Loaded Round Window Niche Implants for an Animal Model using 3D Printed Molds” is an interesting manuscript describing the fabrication process of dexamethasone-loaded implants for the treatment of the inner ear. However, some considerations should be discussed prior to publication: 

• Please remove brands and companies’ names from the abstract. 

• Have the authors consider the possibility of using inkjet printing to fabricate the GPRNI. This would avoid the two-steps process, reducing time and costs. And companies as Stratasys already counts with biocompatible materials for inkjet printers such as Objet. 

• Section 2.2. Why are the authors choosing 100 µm thickness for the first layer? 

• Section 2.3. Please, include a description of how the mixture of the silicone and the DEX is conducted. Was the DEX added solid or in solution? How were the authors sure they were obtaining a homogeneous mixture? 

• Section 3.2. The authors indicate that the injection temperature is 160 °C, higher than the Tg of the polymer used for the mold. Although they state the injection temperature did not affect the mold, did they demonstrate that being above the silicone Tg did not affect the size and shape of the final implant? 

• Section 3.2. The authors have used dexamethasone as drug model for the study of the amount of drug released from the implant. In this case, decomposition temperature of the drug is quite above the injection temperature. But have authors consider the range of drugs that could be used in the future considering this temperature limitation? 

• Section 3.3. The released DEX is still in the range of µm after more than one month of release. Would this dose be enough to achieve a positive pharmacological effect? 

Reviewer 2 Report

This paper was well organized and written. The proposed idea was interesting. The conclusions were fully supported by the experimental results. Minor comments were listed as follows:

1. It would be great if the authors could mark the key dimensions of 3D model in Figure 2B.

2. Why was 3D printing selected to fabricate the molds, not high-precision cutting? Any explanations? 

3. The authors were suggested to include Figures 11 and 12 into one figure.

Minor typos were found in the manuscript. Proofreading was needed for the re-submission. 

Reviewer 3 Report

The article is well written and organized. The goal, methods and results are described in proper manner. The conclusion is backed by results. I have no remarks.

Round 2

Reviewer 1 Report

Thank you for your kind response to the reviewer's comments

Reviewer 2 Report

The authors have already solved all my concerns. Thanks for the efforts.

Minor typos needed to be corrected.